# Degradation of Mitochondria and Oxidative Stress as the Main Mechanism of Toxicity of Pristine Graphene on U87 Glioblastoma Cells and Tumors and HS-5 Cells

**DOI:** 10.3390/ijms20030650

**Published:** 2019-02-02

**Authors:** Sławomir Jaworski, Barbara Strojny, Ewa Sawosz, Mateusz Wierzbicki, Marta Grodzik, Marta Kutwin, Karolina Daniluk, André Chwalibog

**Affiliations:** 1Department of Animal Nutrition and Biotechnology, Faculty of Animal Sciences, Warsaw University of Life Sciences, 02-786 Warsaw, Poland; barbara_strojny@sggw.pl (B.S.); ewa_sawosz@sggw.pl (E.S.); mateusz_wierzbicki@sggw.pl (M.W.); marta_grodzik@sggw.pl (M.G.); marta_kutwin@sggw.pl (M.K.); kdaniluk1@gmail.com (K.D.); 2Department of Veterinary and Animal Sciences, Groennegaardsvej 3, 1870 Frederiksberg, Denmark

**Keywords:** pristine graphene, oxidative stress, mitochondria, apoptosis

## Abstract

Due to the development of nanotechnologies, graphene and graphene-based nanomaterials have attracted immense scientific interest owing to their extraordinary properties. Graphene can be used in many fields, including biomedicine. To date, little is known about the impact graphene may have on human health in the case of intentional exposure. The present study was carried out on U87 glioma cells and non-cancer HS-5 cell lines as in vitro model and U87 tumors cultured on chicken embryo chorioallantoic membrane as in vivo model, on which the effects of pristine graphene platelets (GPs) were evaluated. The investigation consisted of structural analysis of GPs using transmission electron microscopy, Fourier transmission infrared measurements, zeta potential measurements, evaluation of cell morphology, assessment of cell viability, investigation of reactive oxygen species production, and investigation of mitochondrial membrane potential. The toxicity of U87 glioma tumors was evaluated by calculating the weight and volume of tumors and performing analyses of the ultrastructure, histology, and protein expression. The in vitro results indicate that GPs have dose-dependent cytotoxicity via ROS overproduction and depletion of the mitochondrial membrane potential. The mass and volume of tumors were reduced in vivo after injection of GPs. Additionally, the level of apoptotic and necrotic markers increased in GPs-treated tumors.

## 1. Introduction

The discovery of graphene became a new driving force in the development of the nanoindustry. Graphene and graphene-based nanomaterials can be used in many fields of biomedical applications including cancer therapy [1,2], drug/gene delivery [3,4], antimicrobial applications [5], tissue engineering [6], and diagnostics [7]. The atoms of carbon in graphene, bonded with sp2 hybridization, are organized in hexagonal structures, resembling the construction of the honeycomb material. The features which distinguish it from other nanomaterials are: an exceptionally high level of charge and electron mobility, high thermal conductivity and low resistivity [8]. Currently, many forms of graphene have been developed. The forms differ in shape, size, and surface modification, giving them comprehensive physical, chemical, and biological properties. Generally, graphene sheets with small size, sharp edges, and rough surfaces easily internalize into the cell in comparison to larger, smooth sheets [9]. However, the future use of graphene and graphene-based materials in a biological and medical context requires a detailed understanding of these materials, which is necessary to extend their biomedical applications in the future. Several reviews have found that graphene results in various degrees of cell death [10,11,12,13]. 

Glioblastoma grade IV is a primary malignant tumor of the brain which is derived from glial cells [14]. According to the World Health Organization (WHO), it is the highest grade of malignancy (grade IV) [15]. It is also the most common malignant brain tumor, as well as being one of the deadliest human tumors. It is characterized by an intensive cell proliferation, angiogenesis, intensive growth, and penetration into the other tissues [16]. Currently, the standard approach to treating cancer is the maximum safe surgical resection followed by radiation therapy with simultaneous chemotherapy with temozolomide [17]. Despite this broad approach, clinical studies show that the average survival time of the patient is only 15 months [18]. For this reason, alternative therapies, including the use of nanoparticles and nanomaterials, are being examined.

In our previous studies with glioma cell lines treated with graphene oxide (GO) and reduced graphene oxide (rGO), we noted dose-dependent toxicity. Both types of platelets reduced cell viability and proliferation with increasing doses, but rGO was more toxic than GO [12]. The uptake, toxic effects and capability of treatment of graphene were previously studied for U87 cells. It has been reported that the reduced graphene oxide nanoribbons functionalized by amphiphilic polyethylene glycol (rGONR–PEG–RGD) [19] reduced graphene oxide nanomesh (rGONM) [20] and zinc ferrite spinel-graphene [21] exhibited concentration-dependent selective photothermal cyto- and geno-toxic effects of the cells. Furthermore, our previous results with pristine graphene platelets (GPs) demonstrate that the cytotoxicity of GPs on glioma cells increases with increasing GP concentrations from 10 to 100 μg/mL [13]. Graphene caused damage to the plasma membrane and induced apoptosis, thus indicating potential efficacy in brain tumor therapy. In this study, we want to better visualize the changes that occur in cells treated with GPs. We hypothesized that graphene provides the formation of reactive oxygen species (ROS), which are the cause of cell membrane damage and mitochondrial disorders and, finally, cell death. The objectives of this study were to measure the toxicity of GPs and the proapoptotic and necrotic activities of graphene in glioblastoma grade IV cells and non-cancer cells (HS-5) and glioblastoma tumors cultured on chorioallantoic chicken embryo membrane.

## 2. Results

### 2.1. Characterization of Graphene

Figure 1 shows representative transmission electron microscope (TEM) and scanning electron microscope (SEM) images of GPs. Because of their hydrophobic character, GPs are usually visible as many layers, and less often as one. GPs are characterized by irregular, corrugated shape, and sharp edges. The diameter of the platelets ranged from 420 nm to 1.6 μm, but agglomerates were more than 4 μm in diameter. Additionally, dynamic light scattering (DLS) analysis was performed to determine the average hydrodynamic diameter of graphene platelets. Agglomerates ranged between 4.2 to 24 µm.

The zeta potential for all tested concentrations was similar; the mean was -11.5 (Figure 2A). FT-IR spectra of GN is presented in Figure 2B. Band originating from carbon bond (C=C) is seen at 1635 cm^−1^.

### 2.2. Cell Morphology

Microscopic visualization of interactions between control and GPs-treated cells showed that in both cell lines, it was noticeable that GP agglomerates attached to the cell and protrusions (Figure 3). Micrographs of cell cultures exposed to high GP concentrations for 24 h demonstrated cells with altered cell morphology and an increased number of apoptotic cells. The GP-treated cells were more oval, denser, and their protrusions were shorter in comparison with the control cells.

### 2.3. Cell Metabolic Activity

Cell viability was evaluated at 1, 4, 12, and 24 h post-exposure. After 1 h, GPs did not show obvious cytotoxic effects on the U87 and HS-5 cell lines. After a longer exposure period, increased concentration of GPs resulted in decreased vitality in both cell types. The lowest vitality was observed at the GP concentration of 200 μg/mL, with 72% and 54% (after 4 h), 44% and 32% (after 12 h), and 46% and 40% (after 24 h) in U87 and HS-5 cells, respectively (Figure 4).

### 2.4. ROS Production and Mitochondrial Membrane Potential

GPs significantly (*P* < 0.05) increased the ROS production of U87 and HS-5 cells compared with the controls group. Increased concentrations of GPs resulted in increased ROS generation in both cell lines. The highest was observed at a concentration of 200 µg/mL (Figure 5E). The mitochondrial membrane potential is crucial for maintaining the physiological function of the respiratory chain in the production of ATP. A significant loss of ΔΨm causes loss of energy and further death. Non-treated cells have active mitochondria; therefore, they collect aggregates of the orange dye inside them, which are visualized with fluorescence microscopy. The loss of orange fluorescence from the mitochondria indicates the collapse of ΔΨm upon treatment with GPs. Increased concentrations of GPs resulted in an increased ratio of green/orange fluorescence in both cell lines (Figure 5).

### 2.5. Analysis of Macro and Microstructure of U87 Tumors

U87 cells grew successfully on the CAM and were able to rapidly induce the formation of solid tumors ranged from 6 to 12 mm diameter. U87 tumors had an oval shape and well-developed blood vessels on the surface (Figure 6). Blood vessels were clearly visible within the tumor tissue, showing that the U87 glioblastoma tumor cells induced a neovascularization from the chick vasculature. A decrease in tumor mass and volume was observed in the GP-treated group (Figure 6G).

The microstructures in both groups were similar. The surface of the tumor was characterized by a multilamellar flat epithelium, focally keratinizing. There was no significant difference between control and GP-treated tumors in terms of cellular atypia and anaplasia. U87 tumors showed a diffuse pleiomorphic infiltrate of fibrillar and stellate cells with smaller and larger atypical nuclei and a high ratio of nucleus to cytoplasm. Both groups showed high mitotic activity; the mitotic index varied from 6.6 in control tumors to 5.4 in GPs-treated tumors. In the GP treated group, single abnormal mitoses and apoptotic bodies were observed. Tumor necrosis was found in both groups.

### 2.6. TEM Analysis of Glioma Tumors

Figure 7 shows the morphological changes of U87 tumor cells exposed to GPs (500 µg/mL). Cell structures (nucleus, mitochondria, Golgi apparatus, rough endoplasmic reticulum (R.E.R), endocytotic vesicles) were visible in the control group. Most of the cells had a high rate of protein synthesis, which was confirmed by the highly developed R.E.R. Part of the nuclei contained spheroid bodies composed of granular materials. Control cells had oval or rod-shaped mitochondria with a medium or high electron density matrix. The morphology of the glioblastoma cells in the GP-treated group differed from the control group (Figure 7).

The examination of glioblastoma cell ultrastructure revealed that GPs were located inside cells, dispersed in cytosol. GP-treated cells displayed moderate chromatin condensation and cytoplasmic swelling with rupturing of the plasma membrane. We noticed the destruction of mitochondrial structure such as through focal brightening in the matrix, mitochondrial membranes deformation, and mitochondrial swelling.

### 2.7. Protein Levels

Expression of caspase-9 and caspase-3 in the GP-treated tumors increased by 67% and 84%, respectively, compared with the control group. The level of proteins involved in mitochondrial metabolism (mitochondrial respiratory chain complex I–V: NADH dehydrogenase, succinate dehydrogenase, ubiquinol-cytochrome-c reductase, cytochrome c oxidoreductase, and ATP synthase) was significantly lower in GP-treated tumors. A significant increase of the expression level of the protein of the following cytokines: IL-6, IL-8, GM-CSF, GRO (α, β, γ), and MCP-1 was observed (Figure 8).

## 3. Discussion

In our previous study, we demonstrated the cytotoxic effects of GPs on U87 tumor cells in vitro [13]. Based on them, the present study is a follow-up, including both in vitro and in vivo measurements of GP effects. We used two well-defined biological models: an in vitro model (U87 and HS-5 cells) and a chorioallantoic membrane model (CAM) model to determine the effect of the graphene flakes. Graphene platelets were produced by physical methods directly by exfoliation of graphite without the initial stage of oxidation. TEM and SEM images of GPs showed extremely thin structures; however, hydrophobic GPs had poor solubility and created aggregates in salt-containing physiological buffers due to electrostatic charge and nonspecific binding to proteins and lipids. Diameters of the platelets ranged from 420 nm to 1.6 μm, but agglomerates were more than 4 μm in diameter.

Microscopic visualization of the interactions between GPs and treated cells showed that platelets were present on the surface of the body of cells and protrusions (Figure 3). Micrographs of cell cultures exposed to high GPs concentrations for 24 h demonstrated cells with altered cell morphology and an increased number of apoptotic cells. U87 and HS-5 surface after treatment with GPs showed irregularities and laceration. Changes in morphology, shortening of protrusions and microvilli, and adherence of graphene to the surface of cells were also observed in Hep G2 [22,23], THP-1 [24], and PC12 cells [25] after graphene treatment. Probably, the strong hydrophobic interactions of GPs with the cell membrane lipids might have resulted in this accumulation, which eventually led to the deformation of the cell membrane. It has been also reported that the physical trapping the cells by aggregated graphene sheets in the biological media could be one of the effective mechanisms describing cytotoxicity [26].

An evaluation of cell viability showed a toxic influence of GPs on both tested cell lines (Figure 4). After 1 h, GPs did not show obvious cytotoxic effects on the U87 and HS-5 cell lines. After a longer exposure period, increased concentration of GPs resulted in decreased vitality in both cell types. Using a concentration of 200 μg/mL resulted in a survival rate of 46% in U87 cells and 40% in the HS-5 cells. Zhang et al. [25] also demonstrated that graphene layers induce cytotoxic effects, and that these effects are concentration- and shape-dependent. Furthermore, Chatterjee et al. [23] showed that unoxidized graphene is more toxic than GO to Hep G2 cells. Surface modifications of graphene (addition oxygen groups, polyethylene glycol, biopolymers) improved their solubility and significantly reduced toxic interactions with cells and tissues [27,28,29]. The size- and concentration-dependent cyto- and geno-toxicity of the graphene oxide sheets and nanoplatelets were also studied by Akhavan et al. [30]. The graphene flakes with average lateral dimensions of 11 ± 4 nm exhibited a strong potential in the destruction of the human mesenchymal stem cells (hMSCs) with the threshold concentration of 1.0 mg/mL, while the cytotoxicity of the sheets with average lateral dimensions of 3.8 ± 0.4 mm appeared at high concentration of 100 mg/mL after 1 h. Smaller graphene flakes could penetrate into the nucleus of the hMSCs and exhibit some genotoxicity caused by DNA fragmentations and chromosomal aberrations at low concentrations. However, Mendes et al. [31] showed that the larger graphene flakes reduce cell viability as compared to smaller flakes. In addition, the viability reduction correlates with the time and the concentration of the graphene nanoflakes to which the cells are exposed. Moreover, no obvious difference in the uptake was observed between the different sizes of the graphene layers. The physio-chemical action of graphene with the cell membranes is one of the primary causes of GP cytotoxicity. Smaller graphene platelets can relocate to the cytosol because of their small size, sharp edges, and rough surface. GP nanoplatelets were found to pierce through and mechanically disrupt the plasma membrane. In the treated group, we found graphene inside cells. Similar results were also reported by Sasidharan et al. [29], who used confocal microscopy and flow cytometry to observe the accumulation of pristine and functionalized graphene within the cytosol. Furthermore, Akhavan and Ghaderi [32] reported that the cell membrane of the bacteria was effectively damaged by direct contact of the bacteria with the very sharp edges of the nanowalls, resulting in inactivation of the bacteria by the nanowalls. It has been reported that graphene quantum dots [33], nano-sized GO, and pristine graphene [23] caused decreases in the mitochondrial membrane potential. The U87 and HS-5 cells treated with GPs 4 h after exposure showed a marked dose-dependent depolarization in mitochondrial membrane potential (Figure 5). It was observed that exposure to GPs resulted in a decrease in orange fluorescence intensity after JC-10 staining indicating mitochondrial membrane depolarization and a decrease in the number of functional mitochondria. Depolarization of the mitochondrial membrane can be due to the loss of both structural and functional integrity of the mitochondrion [34]. The damage of mitochondrial functions leads to an increase of intracellular ROS formation. Li and co-investigators [35] reported that pristine graphene can induce cytotoxicity through the depletion of the mitochondrial membrane potential and the increase of intracellular ROS, which then triggers apoptosis by activation of the mitochondrial pathway. Disruption of mitochondrial membrane potential could be associated with oxidative disruption of mitochondrial macromolecules such as mitochondrial DNA, lipids, and proteins caused by reaction with intracellular ROS. TEM analysis of GP-treated tumors showed the destruction of mitochondrial structure, focal brightening in the matrix, and mitochondrial membrane disruption and deformation (Figure 7). The level of proteins involved in mitochondrial metabolism (mitochondrial respiratory chain complex I–V) was significantly lower in GP-treated tumors in comparison to the control group. Zhou et al. [36] indicated that exposure of MDA-MB-231 human breast cancer cells, PC3 human prostate cancer cells, and B16F10 mouse melanoma cells to graphene led to the direct inhibition of the electron transfer chain complexes I, II, III, and IV, most likely by disrupting electron transfer between iron-sulfur centers, which is associated with its stronger ability to accept electrons compared to iron-sulfur centers [36]. Bypassing of the mitochondrial electron transport chain by graphene causes inhibition of oxidative phosphorylation and ROS overproduction. The DCFDA–ROS Detection Assay showed a dose-dependent effect of GPs on ROS production. Increased concentrations of GPs resulted in increased ROS generation in both cell lines (Figure 5E). The exact mechanism through which graphene exerts oxidative stress is difficult to identify and still remains to be elucidated for most graphene family materials. ROS regulate several signaling pathways affecting a variety of cellular processes, such as cell metabolism, carcinogenesis, proliferation, migration, differentiation, and cell death [37,38]. Outer mitochondrial membrane permeabilization results translocation of proapoptotic proteins such as cytochrome c, AIF, or Smac/Diablo to cytosol [39]. A loss of cytochrome c from the mitochondria results in a loss of electrons from the electron transport chain and ROS production [40]. Within the cytosol, cytochrome c, together with Apaf-1 and dATP, form the apoptosome complex which activate procaspase-9 [41,42]. Activated caspase-9 becomes available to activate caspase-3. Activation of apoptosis processes was observed in GP-treated U87 tumors, where expression of caspase-9 and caspase-3 was higher by 67% and 84%, respectively. Apoptotic cells were also observed during analysis of morphology and ultrastructure of GP-treated cells and tumors.

Our previous studies showed activation of apoptosis, and also necrosis, in U87 cells after GP treatment [13]. Graphene may initiate inflammatory responses, which are characterized by the production and secretion of proinflammatory cytokines. An analysis of protein levels showed activation of apoptosis and necrosis in U87 GP-treated tumors. A significant increase of the expression level of the protein of the following cytokines: IL-6, IL-8, GM-CSF, GRO (α, β, γ), and MCP-1 was observed (Figure 8). Zhou and co-investigators [43] demonstrated that pristine graphene significantly promotes the secretion of Th1/Th2 cytokines including IL-1α, IL-6, IL-10, TNF-α, and GM-CSF and chemokines such as MCP-1, MIP-1α, MIP-1β, and RANTES, probably by activating TLR-mediated and NF-κB-dependent transcription in macrophages. Furthermore, it has been demonstrated that the structure, surface, and colloidal properties affect the degree of necrosis induction [44,45]. Larger graphene platelets induced inflammation responses that were much stronger in comparison to those with nano-sized flakes [46].

We observed a decrease in tumor growth in weight and volume. In GP-treated tumors, weight decreased by 21% and volume by 31% compared with the control group. We propose that a reduction of mass and volume in treated tumors is associated with the destruction of cells (apoptosis and necrosis) and lower proliferation, which is supported by the mitotic index calculation. The results are comparable with our previous findings using a similar in vivo model, where GO and rGO caused a decrease in U87 tumor volume by 43% and 42%, respectively [12].

## 4. Materials and Methods

### 4.1. Preparation and Characterization of GN

Graphene powder (GPs, purity higher than 99.5%) was purchased from SkySpring Nanomaterials (Houston, TX, USA). The size and shape of the graphene platelets were inspected using a JEM-1220 (JEOL, Tokyo, Japan) TEM at 80 KeV, with a Morada 11 megapixels camera (Olympus Soft Imaging Solutions, Münster, Germany) and FEI QUANTA 200 SEM. The average size of agglomerates and zeta potential measurements were carried out using Zetasizer Nano S90 (Malvern Instruments Ltd., Malvern, UK). using DLS at room temperature (25 °C). Infrared spectra were collected in a Fourier transform infrared spectrophotometer (Nicolet 8700 FTIR, Thermo Scientific, Waltham, MA, USA). Measurements were performed using the FT-IR ATR (attenuated total reflectance) technique over a range of 4000–400 cm^-1^.

### 4.2. Cell Cultures and Treatments

Human glioblastoma U87 cell line and non-cancer cells HS-5 (bone marrow/stroma) obtained from the American Type Culture Collection (Manassas, VA, USA) and maintained in Dulbecco’s modified Eagle’s culture medium containing 10% fetal bovine serum (Life Technologies, Houston, TX, USA), 1% penicillin and streptomycin (Life Technologies) at 37°C in a humidified atmosphere of 5% CO_2_/95% air in a NuAire DH AutoFlow CO_2_ Air-Jacketed Incubator (Plymouth, MN, USA).

### 4.3. Cell Cultures and Treatments

U87 and HS-5 cells were plated in Petri Dishes 35 × 10 mm (1 × 10^5^ cells per well) and incubated for 24 h. Cells cultured in a medium without the addition of GPs were used as the control. Graphene was introduced to the cells at increasing concentrations (20, 50, 100, and 200 μg/mL). Cell morphology was recorded using a holographical microscope Nanolive 3D Cell Explorer with STEVE Software (Nanolive, Ecublens, Switzerland).

### 4.4. Cell Metabolic Activity

Metabolic rate of U87 and HS-5 cells was evaluated using a 2.3-*Bis*-(2-methoxy-4-nitro-5-sulfophenyl)-2H-tetrazolium-5-carboxyanilide salt (XTT)-based cell proliferation assay kit (Merck, Darmstradt, Germany). U87 and HS-5 cells were plated in 96-well plates (5 × 10^4^ cells per well) and incubated for 24 h. Then, the medium was removed, and GPs were introduced to the cells at increasing concentrations (20, 50, 100, and 200 μg/mL). Incubation after the addition of graphene was carried out for 1, 4, 12, and 24 h. In the next step, 50 μL of XTT solution was added to each well and incubated for an additional hour at 37°C. The optical density (OD) of each well was recorded at 450 nm on an enzyme-linked immunosorbent assay reader (Infinite M200, Tecan, Durham, NC, USA). Cell viability was expressed as the percentage of (ODtest–ODblank)/(ODcontrol–ODblank), where “ODtest” is the optical density of cells exposed to GPs, “ODcontrol” is the optical density of the control sample, and “ODblank” is the optical density of wells without cells.

### 4.5. ROS Production

The DCFDA-Cellular Reactive Oxygen Species Detection Assay Kit (Abcam, Cambridge, UK) was used for measurement ROS in U87 and HS-5 cells. U87 and HS-5 cells were plated in 96-well plates (5 × 10^4^ cells per well) and incubated for 24 h. Then, the medium was removed, and GPs were introduced to the cells at increasing concentrations (20, 50, 100, and 200 μg/mL). After 4 h, the medium with graphene was removed and 100 μL of diluted DCFDA was added to each well and incubated for an additional 45 min at 37°C in the dark. ROS production was measured by fluorescence spectroscopy with an excitation wavelength at 485 nm and an emission wavelength at 535 nm on an ELISA reader (Infinite M200, Tecan, Durham, NC, USA).

### 4.6. Mitochondrial Membrane Potential

Mitochondrial membrane potential (Δψm) was evaluated using JC-10 Mitochondrial Membrane Potential Assay Kit (Abcam, Cambridge, UK). Membrane potential is highly interlinked to many mitochondrial processes. U87 and HS-5 cells were plated in 96-well plates (5 × 10^4^ cells per well) and incubated for 24 h. Then, the medium was removed, and GPs were introduced to the cells at increasing concentrations (20, 50, 100, and 200 μg/mL). After 4 h, the medium with graphene was removed and 50 μL of diluted JC-10 was added to each well and incubated for an additional 45 min at 37°C in the dark. Δψm was measured as a ratio of orange and green fluorescence. Fluorescence intensity was monitored at Ex/Em = 490/525 and 540/590 nm on an ELISA reader (Infinite M200, Tecan, Durham, NC, USA) and on fluorescence microscope (Olympus CKX41, Warsaw, Poland).

### 4.7. Culture of GMB on A Chorioallantoic Membrane

Fertilized Ross 308 chicken eggs (*Gallus domesticus*) from a local hatchery were placed in a humidified 37  °C incubator without CO_2_ to induce embryogenesis. After seven days of egg incubation, a silicone ring containing 3 × 10^6^ U87 glioma cells suspended in 20 μL of culture medium was placed on the chorioallantoic membrane (CAM). The eggs were incubated for 10 days then the tumors were resected for further analysis. Eggs were divided into two groups of 45: the control group and GP group (injected with 200 µL of 500 µg/mL solution of GPs). After 3 days, the tumors were resected for further analysis.

### 4.8. Measurement of Tumor Volume

Digital photos of tumors were taken using a stereo microscope (SZX10, CellD software version 3.1; Olympus Corporation, Tokyo, Japan). The measurements were taken with cellSens Dimension Desktop version 1.3 (Olympus). The tumor volumes were calculated with the following equation [47]:V=4/3 πr^3^ where r =1/2 √(diameter 1 × diameter 2), π = 3.1415

### 4.9. Histological Analysis

After resection, tumors were fixed in 10% buffered formaldehyde for 24 h (10% in buffer phosphate). Samples were dehydrated, embedded in paraffin overnight, and cut into sections 5–6 microns thick. Sections were mounted on poly-l-lysine-coated slides (Equimed, Krakow, Poland). The resulting sections were deparaffinized by immersion in two changes of xylene for 10 min each. Sections were then rehydrated in descending series of ethanol ending in water for 5 min. Hematoxilin (Sigma-Aldrich, St. Louis, MO, USA) solution was then applied for 5 min, followed by a final 3 min rinse in water. Eosin solution was applied for 1 min and then dehydration was carried out with 70%, 96%, and 100% alcohol. Finally, the samples were submerged in xylol and mounted. Cells and tissues were measured using a Leica DM750 microscope coupled with a digital camera Leica ICC50 and LAS EZ microscope imaging software (Version 3.0, Leica Microsystems, Wetzlar, Germany). Mitotic index was assessed as the number of mitotic figures in 10 visual fields (40 µm^2^).

### 4.10. TEM Analysis of Tumors

Tumor tissues were cut immediately after resection into pieces of about 1.5 mm^3^ and fixed in a 2.5% glutaraldehyde solution (Sigma-Aldrich) in 0.1 M PBS (pH 7) overnight. The samples were washed in the PBS and transferred to a 1% osmium tetroxide solution (Sigma-Aldrich) in 0.1 M PBS (pH 7) for 1 h, then washed in distilled water, dehydrated in ethanol gradients, and impregnated with epoxy embedding resin (Fluka Epoxy Embedding Medium Kit; Sigma-Aldrich). After 24 h, the samples were embedded in the same resin and baked for 24 h at 36 °C, then transferred to a 60 °C incubator and baked for a further 24 h. The blocks were cut into ultrathin sections (50 nm) using an ultramicrotome (Ultratome III; LKB Products, Vienna, Austria) and transferred onto TEM grids (Formvar on 3 mm 200 Mesh Cu Grids, Agar Scientific, Stansted, UK). Sections were contrasted using uranyl acetate dihydrate (Sigma-Aldrich) and lead (II) citrate tribasic trihydrate (Sigma-Aldrich), and examined by TEM.

### 4.11. Protein Levels

TissueLyser LT (Qiagen, Hilden, Germany) was used to prepare protein extracts. Protein concentrations of tissues lysates were determined using the BCA Protein Assay (Thermo Scientific). The levels of multiple proteins engaged in the inflammatory state, hypersensitivity, and mitochondrial metabolism were evaluated using Western blot analysis, ELISA and membrane arrays. Caspase-3, Caspase-9, and NFĸB levels were examined by Western blot analysis. An equal volume (50 mg) of samples was denatured by the addition of sample buffer (Bio-Rad Laboratories, Munich, Germany) and boiled for 5 min. Proteins from U87 glioma tissues were subjected to SDS/PAGE and then transferred to polyvinylidene difluoride membranes (Life Technologies, Gaithersburg, MD, USA) and probed with primary antibodies anti-Caspase 3 (cat. no. NB100-56708, Novus Biologicals, Centennial, CO, USA), anti-Caspase 9 (cat. no. NB100-56118), with GAPDH (cat. no. NB300-327, Novus Biologicals) as the loading control. After incubation with secondary fluorescent antibodies, the proteins were detected by the GelDoc scanner (Bio-Rad Laboratories, Germany), using the fluorescent method. Protein bands were evaluated using the Quantity One 1-D analysis software (Version 4.6, Bio-Rad Laboratories, Munich, Germany).

The levels of multiple cytokines were examined by the Human Cytokine Antibody Array Membrane (cat. no. ab133996, Life Technologies), prepared for the simultaneous detection of 23 cytokines. The following targets can be detected by this array: G-CSF (granulocyte colony-stimulating factor), GM-CSF (granulocyte macrophage colony-stimulating factor), GRO-α (growth regulated oncogene alpha precursor), IL-1 α (interleukin 1 α), IL-2 (interleukin 2), IL-3 (interleukin 3), IL-5 (interleukin 5), IL-6 (interleukin 6), IL-7 (interleukin 7), IL-8 (interleukin 8), IL-10 (interleukin 10), IL-13 (interleukin 13), IL-15 (interleukin 15), IFN-γ (interferon gamma), MCP-1 (monocyte chemoattractant protein 1), MCP-2 1 (monocyte chemoattractant protein 2), MCP-3 1 (monocyte chemoattractant protein 3), MIG (monokine induced by gamma interferon), RANTES (chemokine ligand 5), TGF-β1(transforming growth factor beta 1), TNF-α (tumor necrosis factor alpha), and TNF-β (tumor necrosis factor beta). Three samples from each group were diluted to a final concentration of 5 μg/μL. The cytokine array was performed according to the instructions. Chemiluminescence detection was performed using multiple exposure times (30 s to 5 min) with the ChemiDoc1 Imaging System with Quantity One Basic Software (Bio-Rad, Hercules, CA, USA).

Proteins involved in mitochondrial metabolism (mitochondrial respiratory chain complex): NADH dehydrogenase (Complex I), succinate dehydrogenase (Complex II), ubiquinol-cytochrome-c reductase (Complex III), cytochrome c oxidoreductase (Complex IV), and ATP synthase (Complex V) were examined using Enzyme-linked immunosorbent assays (Abcam, cat. no. ab178011, cat. no. ab124536, cat. no. ab124537, cat. no. ab179880, cat. no. ab124539). The cytokine array was performed according to the instructions. The intensities of signals were quantified using the ELISA reader.

### 4.12. Statistical Analysis

Data were analyzed using multifactorial and monofactorial analysis of variance with Statgraphics^®^ Plus 4.1 (StatPoint Technologies, Warrenton, VA, USA). The differences between groups were tested using Tukey’s multiple range tests. All mean values are presented with the standard deviation. Differences with *P* < 0.05 were considered significant.

## 5. Conclusions

The in vitro results with U87 glioma cell line and HS-5 normal cells demonstrated that GPs cause a dose-dependent cytotoxicity via ROS overproduction and depletion of the mitochondrial membrane potential. Additionally, the level of apoptotic and necrotic markers increased in GP-treated tumors. The cytotoxic responses were confirmed in vivo after injection of GPs, showing reduced mass and volume of U87 tumor tissue. The results indicate a potential applicability of GPs in tumor therapy, but side-effects on normal cells must be considered further.

## Figures and Tables

**Figure 1 ijms-20-00650-f001:**
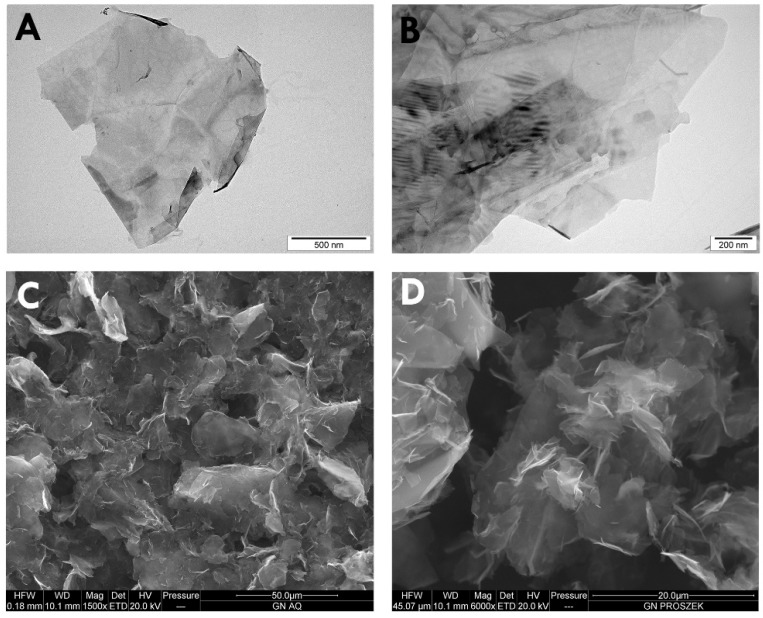
Characterization of pristine graphene by transmission electron microscopy (**A**,**B**) and scanning electron microscopy (**C**,**D**).

**Figure 2 ijms-20-00650-f002:**
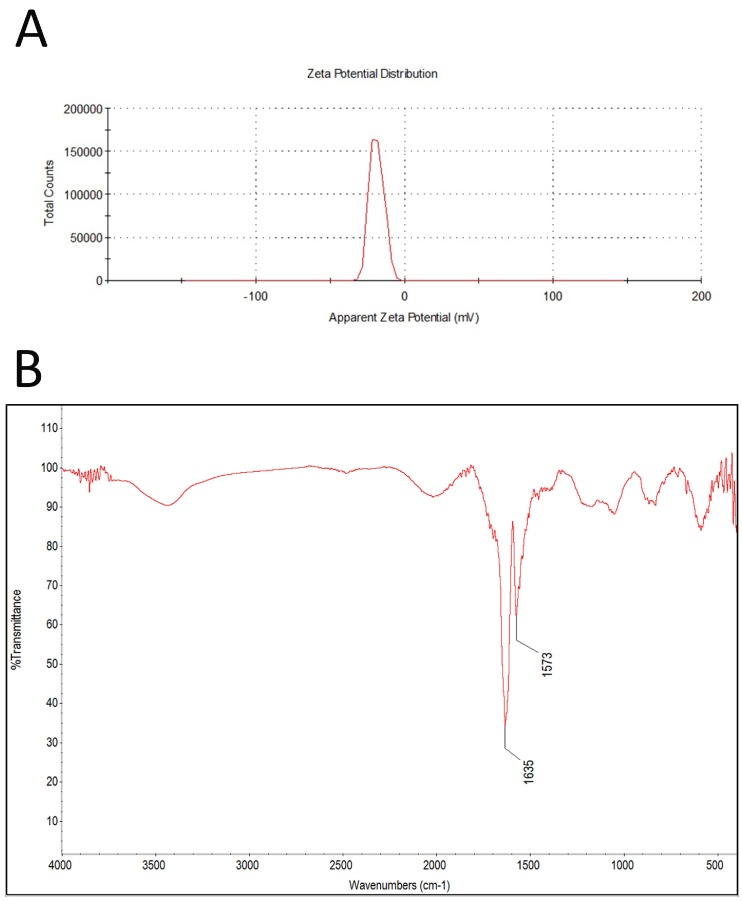
Test results of zeta potential (**A**), and FT-IR (ATR, attenuated total reflectance) spectrum of pristine graphene (**B**).

**Figure 3 ijms-20-00650-f003:**
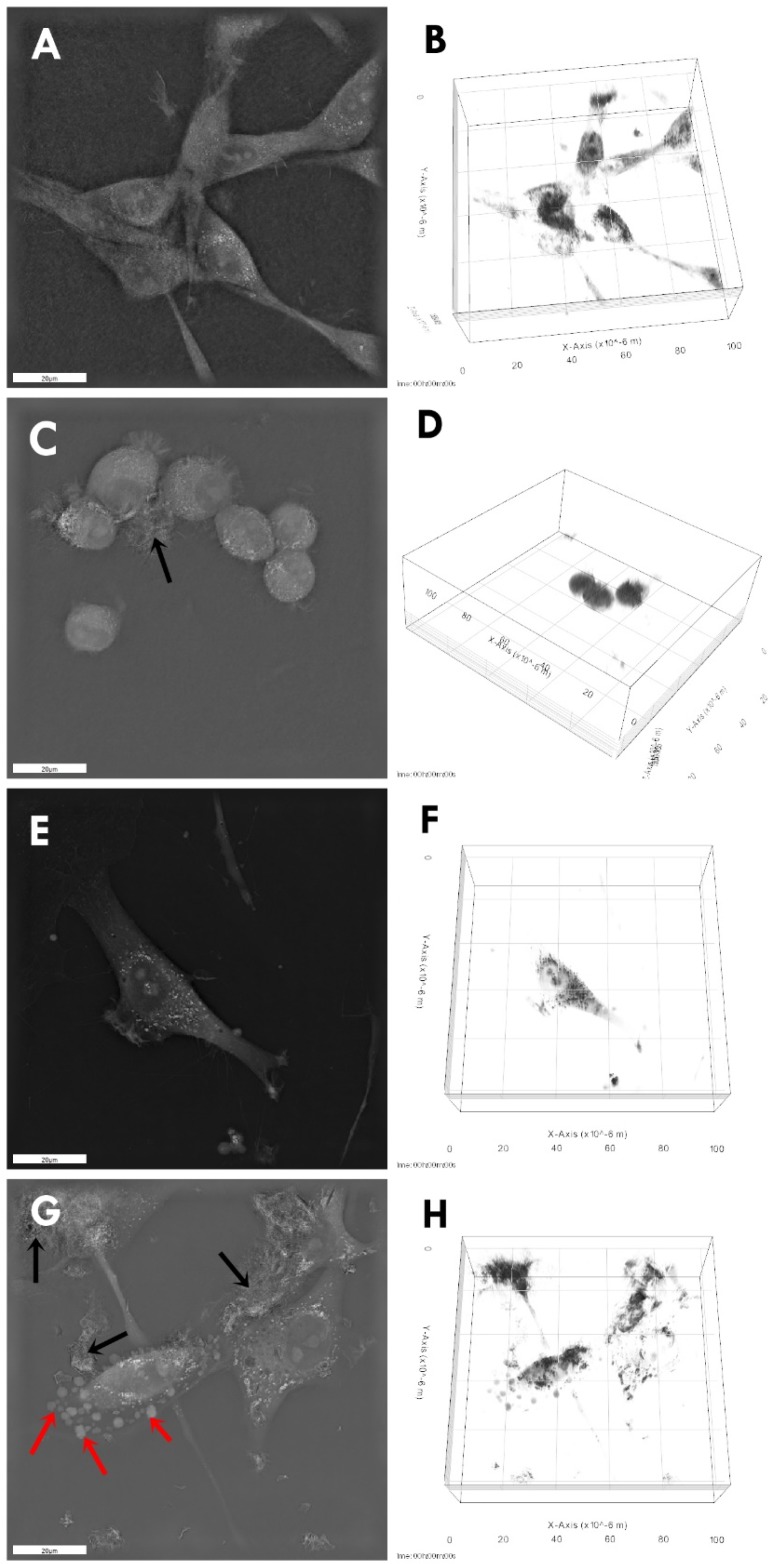
Morphology of U87 (**A**–**D**) and HS-5 (**E**–**H**) cells: untreated control (**A**,**B**,**E**,**F**), treated with pristine graphene (**C**,**D**,**G**,**H**). Notes: Black arrows point to graphene agglomerates. Red arrows point to apoptotic bodies.

**Figure 4 ijms-20-00650-f004:**
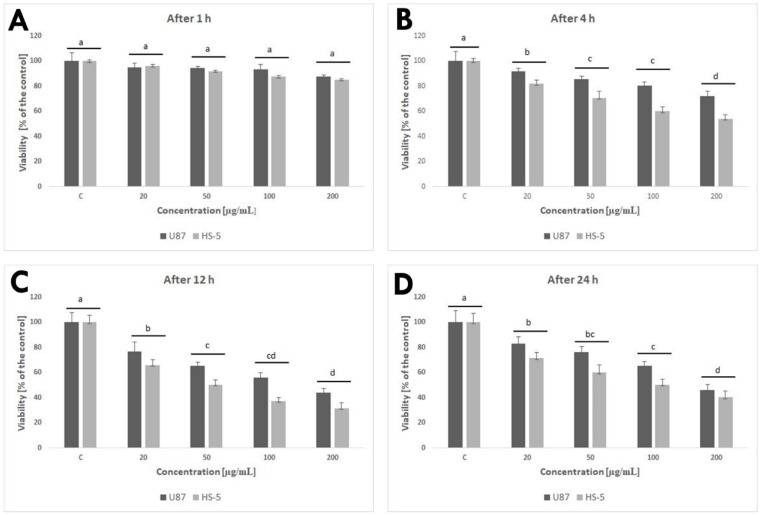
Effect of pristine graphene on the viability of U87 and HS-5 cells after 1 (**A**), 4 (**B**), 12 (**C**), and 24 (**D**) h. Notes: The columns with different letters (a–d) indicate significant differences between the concentrations; error bars are standard deviations. C—control group (untreated cells).

**Figure 5 ijms-20-00650-f005:**
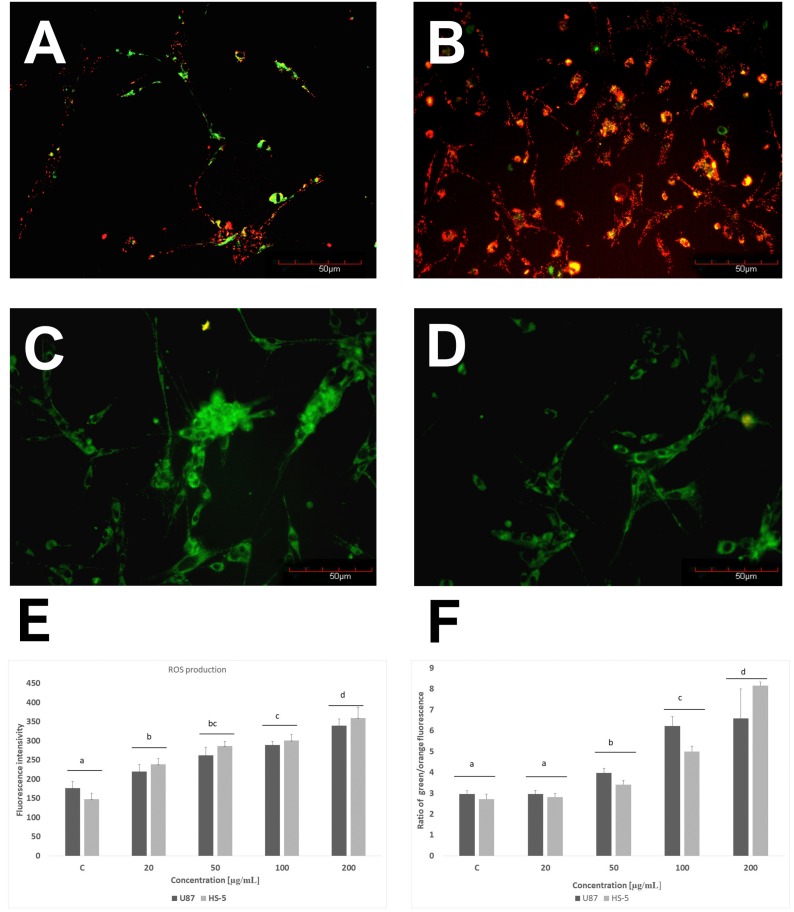
Investigation of mitochondrial transmembrane potential of U87 (**A**,**C**) and HS-5 cells (**B**,**D**) and ROS production (**E**). **A**,**B**–control cells, **C**,**D**–cells exposed to 50 µg/mL of GPs. **F**–ratio of green/orange fluorescence.

**Figure 6 ijms-20-00650-f006:**
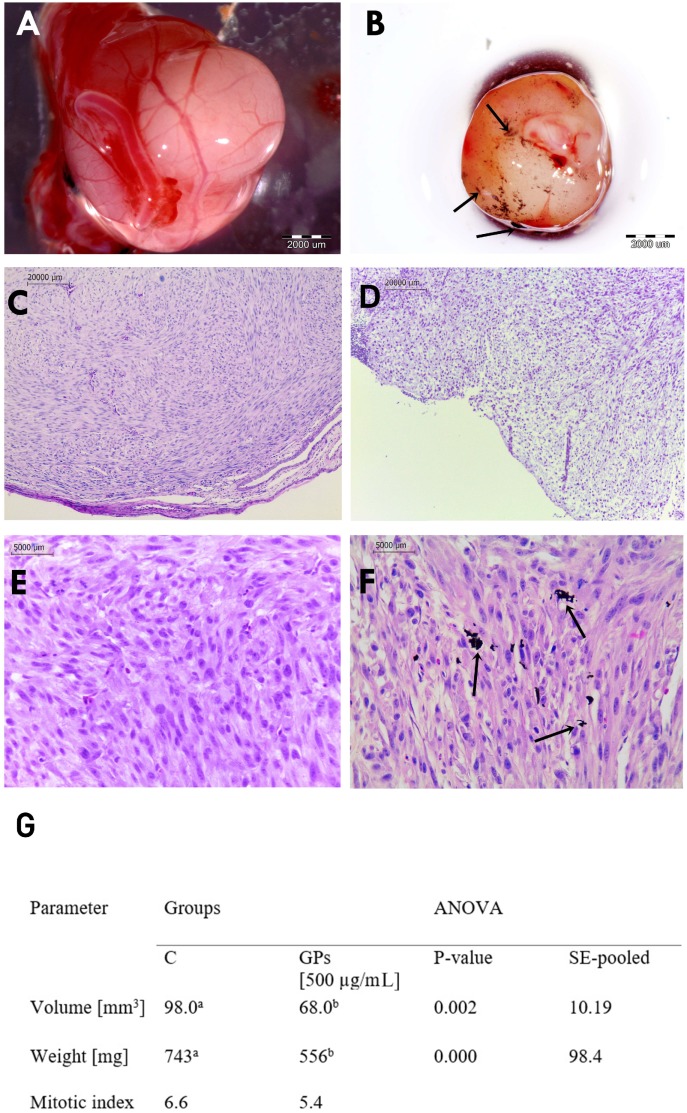
Glioblastoma multiforme tumor cultured on chorioallantoic membrane. (**A**,**C**,**E**) control group; (**B**,**D**,**F**) pristine graphene treated group. (**G**) U87 tumor volume, weight, and mitotic index in the control (**C**) and pristine graphene (GPs) groups. Notes: Black arrows point to graphene agglomerates. The columns with different letters (a–b) indicate significant differences between the groups.

**Figure 7 ijms-20-00650-f007:**
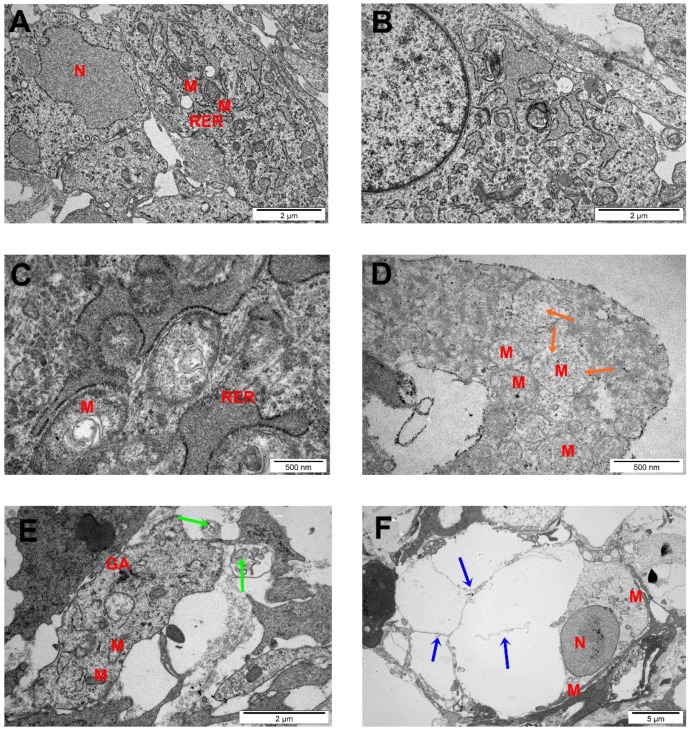
Glioblastoma multiforme tumors ultrastructure from control group (**A**,**B**) after GPs treatment (**C**–**F**). Notes: Scale bar: A, B, E 2 μm; C and D 500 nm; F 2 μm. Green arrows point to graphene agglomerates, orange arrows point to degraded mitochondria, blue arrows point to apoptotic bodies. Abbreviations: N—nucleus, M—mitochondria, RER—rough endoplasmic reticulum, AG—Golgi apparatus.

**Figure 8 ijms-20-00650-f008:**
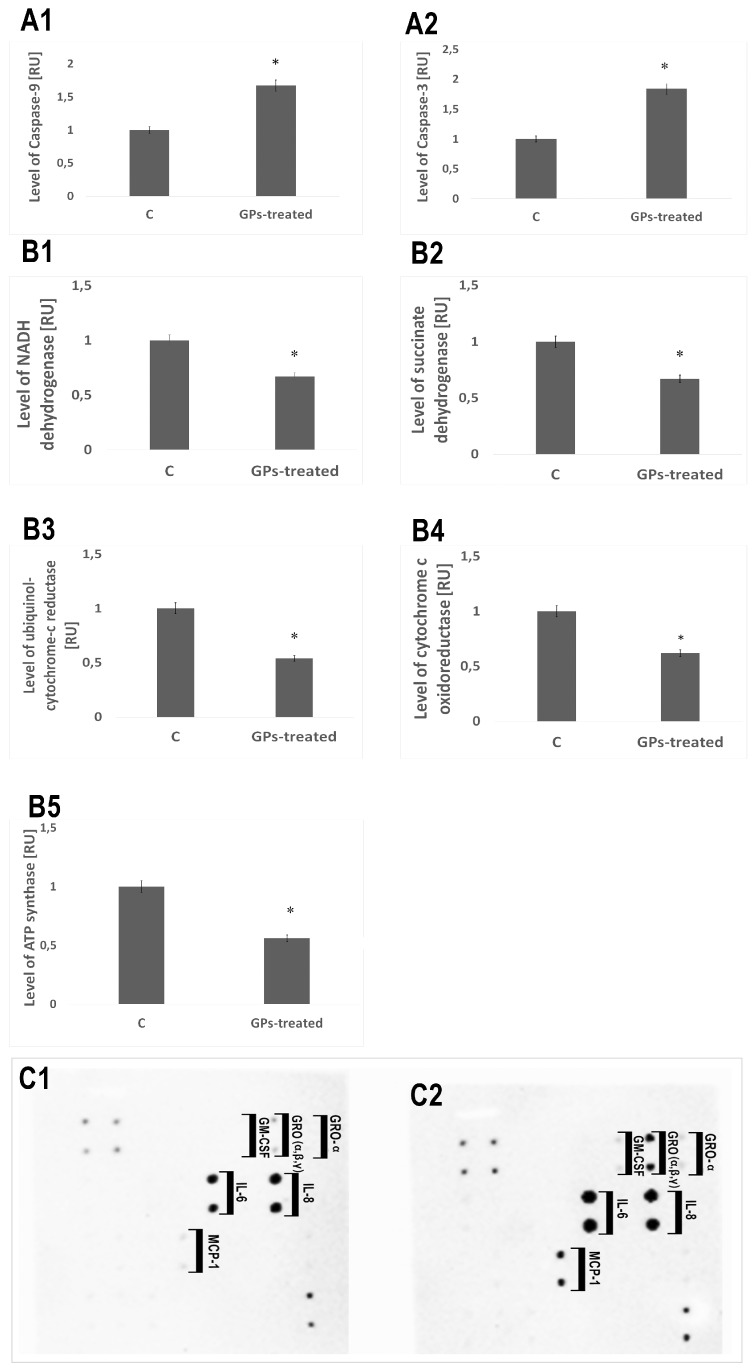
Protein expression level. Expression of caspase-9 and caspase-3 in the pristine graphene treated tumors significantly increased compared with the control group (**A1**,**A2**). Level of proteins involved in mitochondrial metabolism (mitochondrial respiratory chain complex: NADH dehydrogenase, succinate dehydrogenase, ubiquinol-cytochrome-c reductase, cytochrome c oxidoreductase, and ATP synthase) was significantly lower in pristine graphene treated tumors (**B1**–**B5**). A significant increase of the expression level of the protein of the following cytokines: IL-6, IL-8, GM-CSF, GRO (α, β, γ), MCP-1 was observed (**C1**,**C2**). Abbreviations: RU—relative units.

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
