# Peer review of "Degradation of Mitochondria and Oxidative Stress as the Main Mechanism of Toxicity of Pristine Graphene on U87 Glioblastoma Cells and Tumors and HS-5 Cells"

_ijms, 2019, doi:10.3390/ijms20030650_

Round 1
Reviewer 1 Report
In this work, toxicity of graphene on some cells was studied and tried to present a mechanism. The work is potentially interesting and publishable. But, there are some points which should be addressed by the authors before any further consideration. I suggest revision of the manuscript based on the following comments:
1. In Figure 2, FTIR shows some peaks relating to functional groups on the graphene. But, the authors called the graphene sheets “pristine graphene”. By “pristine graphene” one would except a pure carbon structure with honeycomb bonds between carbon atoms in a plane. It seems that the material used by the authors was really graphene oxide rather than “pristine graphene”. Now, this should be clarified by the authors. The same can be stated for surface charge (zeta potential). I suggest the authors compare the zeta potential of the materials used in this work with that of “pristine graphene” and GO in the literature.
2. The uptake, toxic effects and capability of treatment of graphene oxide were previously studied for U87 cell. See, for example [small 2013, 9, No. 21, 3593–3601], [J. Mater. Chem., 2012, 22, 20626–20633], [J. Am. Chem. Soc., 2011, 133, 6825–6831.] and [J. Mater. Chem. B, 2014, 2, 3306–3314]. These should be noted in the Introduction section for further completion of the literature review. In addition, the authors should highlight the advantages/disadvantages of this work as compared to the previous ones.
3. The authors mentioned ROS generation as one of the factor contributed in the toxicity of the graphene sheets. But, there are some known mechanisms which should be considered, discussed or at least noted by the authors. For example, 1) the cells can be damaged by sharp edges of the sheets and loss their cytoplasm. Furthermore, the nuclei of the cells can be damaged by this physical mechanisms, resulting in DNA/RNA damage (see, for example [Biomaterials 33 (2012) 8017-8025] and [ACS Nano VOL. 4 ▪ NO. 10 ▪ 5731–5736 ▪ 2010]). 2) Trapping the cells by aggregated graphene sheets in the biological media (which usually can reduce the oxygen groups on the graphene oxide). See, for example [J. Phys. Chem. B 2011, 115, 6279–6288] and [RSC Adv., 2014, 4, 27213–27223].
Author Response
Dear Editor
Thank you for the most valuable comments from the referees. In the following are our answers. All changes are in red. The original manuscript was revised by the Proof-Reading-Service.com and we carefully checked it again. All authors agree to the changes of the manuscript.
Reviewer 1 Evaluation:
1. In Figure 2, FTIR shows some peaks relating to functional groups on the graphene. But, the authors called the graphene sheets “pristine graphene”. By “pristine graphene” one would except a pure carbon structure with honeycomb bonds between carbon atoms in a plane. It seems that the material used by the authors was really graphene oxide rather than “pristine graphene”. Now, this should be clarified by the authors. The same can be stated for surface charge (zeta potential). I suggest the authors compare the zeta potential of the materials used in this work with that of “pristine graphene” and GO in the literature.
Thank you for valuable coment. Graphene powder (GPs, purity higher than 99.5%) was purchased from SkySpring Nanomaterials (Houston, USA). This Graphene is described as pristine Graphene
(https://ssnano.com/inc/sdetail/graphene_nanopowder/3368)
We've carried out the FT-IR analysis again and the FT-IR spectrum that we have receivedis different from the previous one and typical for the pristine graphene. The previous one had to be a mistake. The zeta potential for all tested concentrations was similar, the mean was -11.5. This is a low value indicating graphene instability, which is confirmed by literature data. Test results of zeta potential, and FT-IR (ATR, attenuated total reflectance) spectrum of pristine graphene was corrected as below:
Results:
2.1. Characterization of graphene
The zeta potential for all tested concentrations was similar, the mean was -11.5 (Figure 2A). FT-IR spectra of GN is presented in Figure 2B. Band originating from carbon bond (C=C) is seen at 1635 cm-1.
Figure 2. Test results of zeta potential (A), and FT-IR (ATR, attenuated total reflectance) spectrum of pristine graphene (B).
2. The uptake, toxic effects and capability of treatment of graphene oxide were previously studied for U87 cell. See, for example [small 2013, 9, No. 21, 3593–3601], [J. Mater. Chem., 2012, 22, 20626–20633], [J. Am. Chem. Soc., 2011, 133, 6825–6831.] and [J. Mater. Chem. B, 2014, 2, 3306–3314]. These should be noted in the Introduction section for further completion of the literature review. In addition, the authors should highlight the advantages/disadvantages of this work as compared to the previous ones.
New information has been added in introduction section:
“In our previous studies with glioma cell lines treated with graphene oxide (GO) and reduced graphene oxide (rGO) we noted dose-dependent toxicity. Both types of platelets reduced cell viability and proliferation with increasing doses, but rGO was more toxic than GO [12]. The uptake, toxic effects and capability of treatment of graphene were previously studied for U87 cells. It has been reported that the reduced graphene oxide nanoribbons functionalized by amphiphilic polyethylene glycol (rGONR–PEG–RGD) [19] reduced graphene oxide nanomesh (rGONM) [20] and zinc ferrite spinel-graphene [21] exhibited concentration-dependent selective photothermal cyto- and geno-toxic effects of the cells. Furthermore, ourprevious results with pristine graphene platelets (GPs) demonstrate that cytotoxicity of GPs on glioma cells increases with increasing GP concentrations from 10 to 100 μg/mL [13].
3. The authors mentioned ROS generation as one of the factor contributed in the toxicity of the graphene sheets. But, there are some known mechanisms which should be considered, discussed or at least noted by the authors. For example, 1) the cells can be damaged by sharp edges of the sheets and loss their cytoplasm. Furthermore, the nuclei of the cells can be damaged by this physical mechanisms, resulting in DNA/RNA damage (see, for example [Biomaterials 33 (2012) 8017-8025] and [ACS Nano VOL. 4 ▪ NO. 10 ▪ 5731–5736 ▪ 2010]). 2) Trapping the cells by aggregated graphene sheets in the biological media (which usually can reduce the oxygen groups on the graphene oxide). See, for example [J. Phys. Chem. B 2011, 115, 6279–6288] and [RSC Adv., 2014, 4, 27213–27223].
New information about mechanisms has been supported in discussion:
“Probably, the strong hydrophobic interactions of GPs with the cell membrane lipids might have resulted in this accumulation, which eventually led to the deformation of the cell membrane. It has been also reported that the physical trapping the cells by aggregated graphene sheets in the biological media could be one of the effective mechanisms describing cytotoxicity [26].”
and
“The size- and concentration-dependent cyto- and geno-toxicity of the graphene oxide sheets and nanoplatelets were also studied by Akhavan et al. [32]. The graphene flakes with average lateral dimensions of 11 ± 4 nm exhibited a strong potential in destruction of the human mesenchymal stem cells (hMSCs ) with the threshold concentration of 1.0 mg/mL, while the cytotoxicity of the sheets with average lateral dimensions of 3.8 ± 0.4 mm appeared at high concentration of 100 mg/mL after 1 h. Smaller graphene flakes could penetrate into the nucleus of the hMSCs and exhibit some genotoxicity caused to DNA fragmentations and chromosomal aberrations at low concentrations. However, Mendes et al. [33] showed that the larger graphene flakes reduce cell viability as compared to smaller flakes. In addition, the viability reduction correlates with the time and the concentration of the graphene nanoflakes to which the cells are exposed. Moreover, no obvious difference in the uptake was observed between the different sizes of the graphene layers.”
and
“GP nanoplatelets were found to pierce through and mechanically disrupt the plasma membrane. In the treated group, we found graphene inside cells. Similar results were also reported by Sasidharan at al. [29], who used confocal microscopy and flow cytometry to observe the accumulation of pristine and functionalized graphene within the cytosol. Furthermore, Akhavan and Ghaderi [34] presented that the cell membrane of the bacteria was effectively damaged by direct contact of the bacteria with the very sharp edges of the nanowalls, resulting in inactivation of the bacteria by the nanowalls.It has been reported that graphene quantum dots [30], nano-sized GO, and pristine graphene [23] caused decreases in the mitochondrial membrane potential.”

Reviewer 2 Report
Review attached.

Author Response
Dear Editor
Thank you for the most valuable comments from the referees. In the following are our answers. All changes are in red. The original manuscript was revised by the Proof-Reading-Service.com and we carefully checked it again. All authors agree to the changes of the manuscript.
Reviewer 2 Evaluation:
1) Please read the paper carefully, for English language style and spelling, and make appropriate corrections and changes. In particular, authors should revise carefully all the Abbreviations in whole text.
The original manuscript was revised by the Proof-Reading-Service.com and we carefully checked it again. All authors agree to the changes of the manuscript.
2) As authors mentioned in ‘Introduction’ section, many forms of graphene have been developed. If so, please describe why authors chose a GP, instead of other types of graphene derivatives.
The authors tested several types of graphene in previous studies. The results indicated that the most toxic was GPs. The explanation of the choice of GN is included in introduction section
„In our previous studies with glioma cell lines treated with graphene oxide (GO) and reduced graphene oxide (rGO) we noted dose-dependent toxicity. Both types of platelets reduced cell viability and proliferation with increasing doses, but rGO was more toxic than GO [12]. … Furthermore, our previous results with pristine graphene platelets (GPs) demonstrate that cytotoxicity of GPs on glioma cells increases with increasing GP concentrations from 10 to 100 μg/mL [13]. Graphene caused damage to the plasma membrane and induced apoptosis, thus indicating potential efficacy in brain tumor therapy. In this study, we want to better visualize the changes that occur in cells treated with GPs.”
3) It has been extensively acknowledged that the size of graphene nanomaterials is a very important factor that directly affect their cytotoxic effects. In the present study, the GPs have various diameters ranged from 420 nm to 1.6 μm. If so, the manuscript should also include some discussion about the size-dependent cytotoxic effects of graphene.
Discussion about size-dependent cytotoxic effect of graphene has been supported:
“The size- and concentration-dependent cyto- and geno-toxicity of the graphene oxide sheets and nanoplatelets were also studied by Akhavan et al. [32]. The graphene flakes with average lateral dimensions of 11 ± 4 nm exhibited a strong potential in destruction of the human mesenchymal stem cells (hMSCs ) with the threshold concentration of 1.0 mg/mL, while the cytotoxicity of the sheets with average lateral dimensions of 3.8 ± 0.4 mm appeared at high concentration of 100 mg/mL after 1 h. Smaller graphene flakes could penetrate into the nucleus of the hMSCs and exhibit some genotoxicity caused by DNA fragmentations and chromosomal aberrations at low concentrations. However, Mendes et al. [33] showed that the larger graphene flakes reduce cell viability as compared to smaller flakes. In addition, the viability reduction correlates with the time and the concentration of the graphene nanoflakes to which the cells are exposed. Moreover, no obvious difference in the uptake was observed between the different sizes of the graphene layers.”
4) Regarding Figure 3 legend, (A, B, D, F) should be (A, B, E, F).
Figure 3 legend has been modified:
„Figure 3. Morphology of U87 (A-D) and HS-5 (E-H) cells: untreated control (A, B,E, F), treated with pristine graphene (C, D, G, H). Notes: Black arrows point to graphene agglomerates. Red arrows point to apoptotic bodies.”
5) Why cell viability was tested for 24 hours and not later?
The chemical exposure duration ranged from 5 minutes to 6 weeks, but most frequently was 24 hours. Cell viability was evaluated 1, 4, 12, and 24 hours post-exposure. The chosen time points in the XXT assay made it possible to observe the toxicity of graphene and enabled the choice of incubation time for further analysis determining the mechanism of toxicity. Theoretically, we could have tested viability longer for example 48h or 72h but we wanted to observe the first changes in cells after graphene treatment and not the derivative effects.
6) Regarding Figure 6, please specify the concentration of GP.
Information about concentration of GPs has been added as below:
7) Regarding TEM analysis of glioma tumors, U87 tumor cells were exposed to 500 μg/mL of GPs. Why did authors choose the concentration?
We used concentation 500 μg/mL of GPs for all in vivo experiments (not only for TEM analysis). It was presented in the Material and Methods section:
„The eggs were incubated for 10 days then the tumors were resected for further analysis. Eggs were divided into two groups of 45: the control group and GP group (injected with 200 µL of 500 µg/ml solution of GPs). After 3 days, the tumors were resected for further analysis.”
We based on our previous studies with carbon nanoparticles (Long term influence of carbon nanoparticles on health and liver status in rats doi: 10.1371/journal.pone.0144821, In vitro and in vivo effects of graphene oxide and reduced graphene oxide on glioblastoma doi:10.2147/IJN.S77591 andNanoparticles of carbon allotropes inhibit glioblastoma multiforme angiogenesis in ovo doi: 10.2147/IJN.S25528).We chose the higher concentration of graphene for injecting the tumors than for the cell treatment, however, not too high.
8) Regarding Figure 7, there is no Figure 7G. In addition, please describe what orange/green/blue arrows indicate.
It was a mistake. Figure 7 legend has been corrected as below:
Figure 7. Glioblastoma multiforme tumors ultrastructure from control group (A, B) after GPs treatment (C-F). Notes: Scale bar: A, B, E 2 μm; C and D 500 nm; F 2 μm. Green arrows point to graphene agglomerates, orange arrows point to degraded mitochondria, blue arrows point to apoptotic bodies. Abbreviations: N – nucleus, M – mitochondria, RER – rough endoplasmic reticulum, AG – Golgi apparatus.
9) As authors mentioned in ‘Discussion’ section, GPs may agglomerate in biological or physiological buffers, leading to higher hydrodynamic diameters. Therefore, I strongly recommend that authors try to conduct size analysis for determining the hydrodynamic size of GPs using dynamic light scattering (DLS) technique.
Analysis of the size of agglomerates using dynamic light scattering (DLS) technique has been supported as below:
4. Materials and Methods
4.1. Preparation and characterization of GN
The average size of agglomerates and zeta potential measurements were carried out using Zetasizer Nano S90 (Malvern Instruments Ltd., Malvern, UK). using DLS at room temperature (25 °C).
and
2. Results
2.1. Characterization of graphene
Additionally, dynamic light scattering (DLS) analysis was performed to determine the average hydrodynamic diameter of graphene platelets. Agglomerates ranged between 4,2 µm to 24 µm.
However, the chart has not been added, because in the case of large agglomerates, the computer program gives only values, not a chart:
10) Authors mentioned that “GP nanoplatelets were found to pierce through and mechanically disrupt the plasma membrane.” in ‘Discussion’ section (page 10). However, their data are not enough to explain it, unfortunately. To investigate appropriate membrane integrity assessment related to disruption of plasma membrane would be required. It would be better to show how GPs can pierce through and mechanically disrupt the plasma membrane.
Membrane integrity was evaluated in our previous publication: In vitro evaluation of the effects of graphene platelets on glioblastoma multiforme cells. Int J Nanomedicine. 2013; 8: 413–420.doi: 10.2147/IJN.S39456. GPs disrupted cell membrane functionality and integrity and there were significant differences between cell lines and GP concentrations.
We mentioned this in the introduction section: “Furthermore, the previous results with pristine graphene platelets (GPs) demonstrate that cytotoxicity of GPs on glioma cells increases with increasing GP concentrations from 10 to 100 μg/mL [13]. Graphene caused damage to the plasma membrane and induced apoptosis, thus indicating potential efficacy in brain tumor therapy. In this study, we want to better visualize the changes that occur in cells treated with GPs.”
11) Authors mentioned that “The results indicate a potential applicability of GPs in tumor therapy.” in ‘Conclusions’ section (page 14). However, considering that the GPs exhibited cytotoxic effects on not only U87 glioma cells, but also non-cancer HS-5 cell lines, the tumor-specific toxicity was not demonstrated. The experimental sets are extremely limited to conclude any finding. Please discuss it.
Yes, GPs exhibited cytotoxic effects on not only U87 glioma cells, but also non-cancer HS-5 cell lines. However, chemotherapeutics also do not show selective toxicity. Considerable frequency and intensity of side effects to bone marrow and liver was noticed after temozolomide (popular in glioma therapy) treatment. Also, administration of graphene is not clear. It seems that the injection directly into the tumor could be the best method. This will reduce the toxicity of graphene to the place of administration.
The conclusion is modified:
The results indicate a potential applicability of GPs in tumor therapy but side-effects on normal cells must be considered further.

Round 2
Reviewer 2 Report
The authors sincerely provided replies to some critical issues raised in the previous review stage. It is considered that the present version of the manuscript was sufficiently well revised according to the reviewer's comments. This manuscript would be acceptable, unless otherwise explained.